# Fungal community composition predicts forest carbon storage at a continental scale

Mark A. Anthony [1,2,3] ✉, Leho Tedersoo [4], Bruno De Vos[5], Luc Croisé[6], Henning Meesenburg [7], Markus Wagner[7], Henning Andreae[8], Frank Jacob[8], Paweł Lech [9], Anna Kowalska[9], Martin Greve[10], Genoveva Popova[11], Beat Frey [2], Arthur Gessler [1,2], Marcus Schaub [2], Marco Ferretti[2], Peter Waldner[2], Vicent Calatayud [12], Roberto Canullo [13], Giancarlo Papitto[14], Aleksander Marinšek[15], Morten Ingerslev[16], Lars Vesterdal [16], Pasi Rautio [17], Helge Meissner[18], Volkmar Timmermann[19], Mike Dettwiler[1], Nadine Eickenscheidt[20], Andreas Schmitz [20,21], Nina Van Tiel [1,22], Thomas W. Crowther [1] & Colin Averill[1]

Forest soils harbor hyper-diverse microbial communities which fundamentally regulate carbon and nutrient cycling across the globe. Directly testing hypotheses on how microbiome diversity is linked to forest carbon storage has been difficult, due to a lack of paired data on microbiome diversity and in situ observations of forest carbon accumulation and storage. Here, we investigated the relationship between soil microbiomes and forest carbon across 238 forest inventory plots spanning 15 European countries. We show that the composition and diversity of fungal, but not bacterial, species is tightly coupled to both forest biotic conditions and a seven-fold variation in tree growth rates and biomass carbon stocks when controlling for the effects of dominant tree type, climate, and other environmental factors. This linkage is particularly strong for symbiotic endophytic and ectomycorrhizal fungi known to directly facilitate tree growth. Since tree growth rates in this system are closely and positively correlated with belowground soil carbon stocks, we conclude that fungal composition is a strong predictor of overall forest carbon storage across the European continent.

Forests are home to roughly 80% of terrestrial biodiversity[1] and represent one of the world's largest carbon sinks[2–4]. Perhaps the least understood and most complex component of forest biodiversity is the soil microbiome. With a growing need to offset the effects of climate change, there is a rising interest to discover how the biodiversity of Earth's most diverse lifeforms – microbes[5] – affects terrestrial carbon storage[6–10]. Soil microbes mediate unique aspects of the forest carbon cycle. Microbial life is responsible for over 50% of soil respiration[11], most plant litter decomposition[8], and steers tree growth and death via mutualisms and pathogen infections[12–14]. Soil microbial community composition is a measure of the identity and relative abundance of microbial species within communities. While it is well known that the

composition of tree species strongly impacts forest processes such as growth[15], albedo[16], and carbon sequestration[17], a comparable effort to understand how soil microbial community composition impacts whole forest-scale processes is urgently needed.

A growing body of experimental and observational studies suggest that microbial composition can affect entire forest functioning by influencing key forest carbon pools, fluxes, and process efficiencies. For example, dark septate fungal endophytes, an ubiquitous group of biotrophic plant root symbionts, stimulate plant growth 52-138% depending on plant and fungal species[18], and similar observations have been made for different rhizosphere bacterial[19], endophytic bacterial[20], and ecto- and arbuscular mycorrhizal fungal species[21].

Belowground, bacterial, and fungal diversity promotes soil respiration[10], microbial carbon use efficiency[22], and overall decomposition rates[23]. Some of these experimental discoveries also seem to generalize to carbon cycle outcomes in actual forest systems. For example, differences in ectomycorrhizal fungal composition have been linked to a three-fold variation in tree growth rates across Europe[7], an observation consistent with decades of mesocosm experiments[24–26]. Studies on soil biogeochemistry have observed that variation in decomposition rates are linked to differences in bacterial composition[23,27,28] and fungal richness[29], and that these different community types likely explain important variation in soil organic carbon storage[30,31]. All these signatures highlight the potential importance of microbial biodiversity, but how they translate to total forest carbon storage remains unknown.

Globally, forest tree biomass and soil organic carbon represent most of the total forest carbon stock[2], and these two pools may positively[32,33] or negatively[34,35] interact. Soil organic carbon storage is balanced by inputs and outputs, with most carbon inputs derived from net primary production[36]. For this reason, plant growth and soil organic carbon stocks are positively linked in commonly used carbon models such as RothC[37] and CENTURY[38]. Yet, experimental research suggests that this connection is often more complex due to priming[39], mineralogy[40], forest management and disturbance[32], soil carbon saturation[41], and mycorrhizal symbiosis[42]. A recent meta-analysis found that the positive effects of elevated $CO_2$ on plant growth are negatively correlated with changes in soil organic carbon stocks across the globe[35]. This was attributed to enhanced nutrient scavenging from organic matter by ectomycorrhizal fungi that can boost plant growth. Many ectomycorrhizal fungi, but not all, also decay soil organic matter to mine for nitrogen[43], a function that is expected to increase soil organic carbon stocks under nitrogen-limiting conditions[42]. Therefore, due to the context-dependency of plant biomass-soil carbon storage relationships, the link between forest soil microbiomes (i.e., a community of microorganisms) and total forest carbon storage requires explicit examination of both above- and belowground carbon pools.

In this study, we explored how in situ forest properties and processes are linked to soil microbiome composition at a large spatial scale across Europe. We then modeled the extent to which features of the soil microbiome are correlated to forest carbon accumulation and storage, both above- and belowground, and we identified which constituents of the microbiome explain these patterns. Until now, efforts to link soil microbial composition to the major components of forest carbon storage have been limited by a lack of paired data on microbial composition, tree biomass carbon stocks, tree growth, and soil organic carbon stocks. We used DNA sequencing to generate soil microbiome profiles of bacteria and fungi across 238 forest monitoring plots spanning 15 European countries (Fig. 1a). All forest monitoring plots are part of the International Cooperative Programme on Assessment and Monitoring of Air Pollution Effects on Forests (ICP Forests) network and have extensive data on forest carbon cycling and storage above and belowground. This microbiome survey allowed us to generate a unique analysis between forest microbiome profiles and paired, co-located observations of total forest carbon balance signatures at a continental scale. We show that soil fungal communities, especially tree-associated ectomycorrhizal and endophytic guilds, are strong predictors of forest tree growth and biomass. Because tree growth is also positively correlated with soil carbon stocks in this system, fungal composition and diversity are prominent bioindicators of overall forest carbon storage.

## Results and discussion
### Variation in forest biotic conditions is linked to fungal versus bacterial composition
Forest biotic variables were correlated with fungal versus bacterial community composition (Fig. 1b–e). Fungal composition was specifically correlated with the dominant tree type, forest age, and tree growth rate (Fig. 1b, d, f). Since tree growth and tree biomass stocks were also positively correlated themselves ($r = 0.7$, $P < 0.001$), we only included tree growth to avoid issues of co-linearity and redundancy. Conversely, neither forest age nor tree growth rate were correlated with bacterial community composition (Fig. 1f). While patterns were similar between soil horizons for both groups, there was a stronger effect of dominant tree type and forest age on fungal composition in the organic versus mineral soil horizon, while variation in tree growth rate was more tightly linked to mineral than organic horizon fungal community composition. Fungal communities typically differ between broadleaves and conifers, especially in the organic horizon[44,45], and fungal composition specifically varies with forest age[46–48]. While we observed the same dissimilarly patterns in fungal communities with forest type and age, fungal composition varied even more strongly with tree growth rate. It is possible that variation across specific tree species would reveal even finer-scale differences, but exploring this was beyond the goal of our study since most sites were dominated by a single species. Fungal, not bacterial, composition is therefore a uniquely informative marker of forest productivity in addition to forest type and age.

Soil and remaining geographic variables captured similar variation in bacterial and fungal community compositions. Soil clay content, soil pH, and soil carbon stocks were significantly correlated with both fungal and bacterial community compositions, though correlations to soil pH and carbon stocks were stronger for bacteria than fungi (Fig. 1f). For both groups, correlations with soil pH and carbon stocks were twice as strong in the organic compared to mineral soil horizon. These are expected results since soil pH and soil carbon content often co-vary with microbial composition[46,49–55]. However, few microbiome studies measure soil carbon *stocks* - an actual metric of forest carbon storage because carbon content alone does not account for the quantity of soil in a system. We suggest that previous focuses on carbon content have obscured the link between microbial composition and soil carbon storage because carbon content was only weakly correlated with microbiome composition in our study (Supplementary Fig. 1). The remaining geographic characteristics were not tightly linked to microbial composition, except for mean annual precipitation and geographic space. Total variation explained in microbiome composition by all environmental variables based on distance-based redundancy analysis ranged between 22.8–28.2%, the typical amount of variation explained in microbiome composition at large spatial scales[56–58].

### Digging deeper: using microbiome metrics to predict forest tree growth and biomass
Our analyses of microbial community composition indicate that fungal and bacterial communities are differentially linked to multiple metrics of forest carbon storage. Since forest carbon pools and microbial communities interact, this cause-and-effect conundrum could not be resolved in our observational study. But we could investigate these linkages deeper to evaluate which components of the microbial community best captured variation in above and belowground carbon storage. To do this, we also took into consideration known factors affecting forest tree growth and biomass across the ICP Forest network where this work was conducted. Earlier research showed that the best non-microbial predictors of tree growth include nitrogen deposition, stand age, and multiple aspects of climate (see ref. 59). Here, we show that none of these non-microbial predictor variables are strongly multicollinear with microbiome composition and diversity (variance inflation values ≤ 5 in all models *sensu*[60]), but they are important predictors of tree growth in our study (see supplementary tables and datasets referenced throughout the results section). Our analyses of the environmental predictors of microbial composition also demonstrate that non-microbial predictors of tree growth are only partially

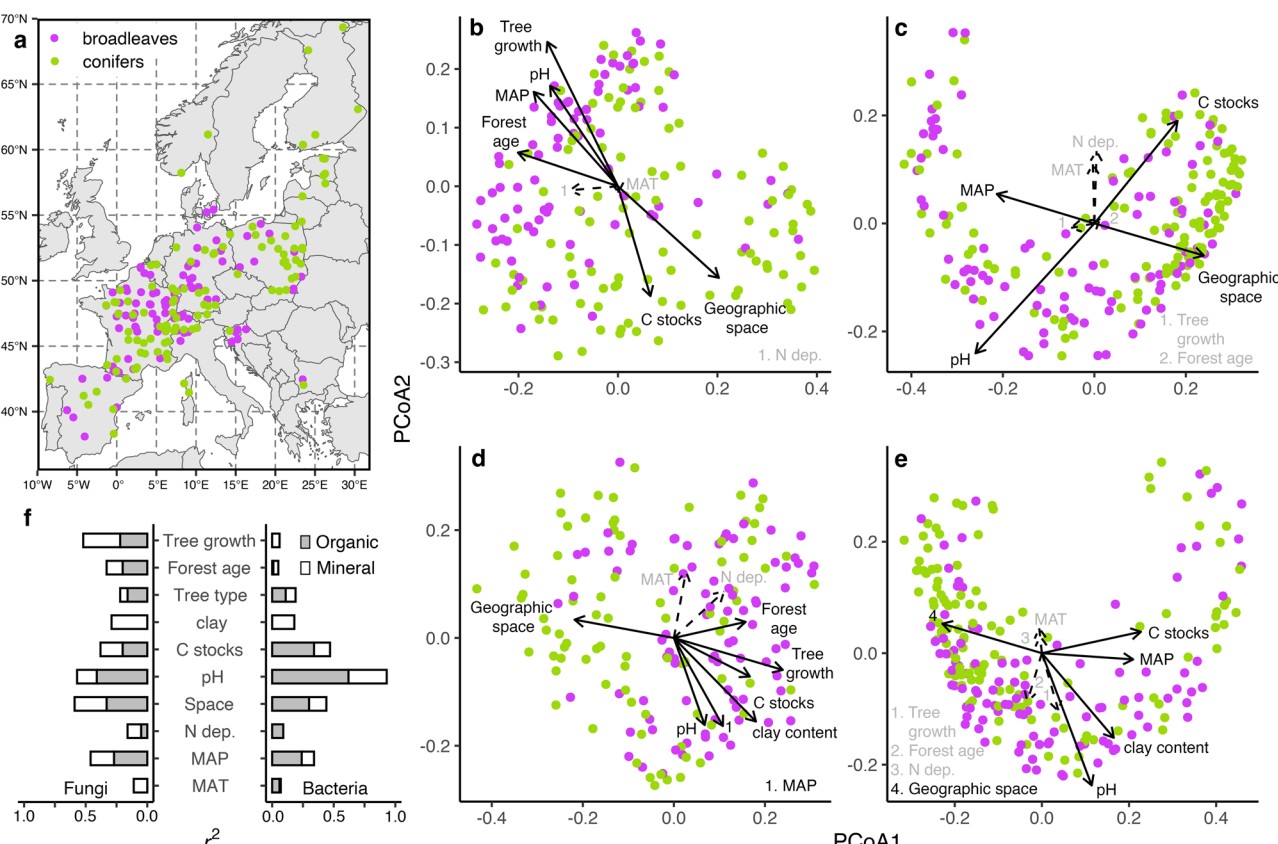

**Fig. 1 | Relationships between the soil microbiome and forest abiotic and biotic conditions.** Soil was collected from 285 forest monitoring plots across 15 European countries participating in the International Co-operative Programme on Assessment and Monitoring of Air Pollution Effects on Forests (ICP Forests) network (**a**). Forests were classified as either broadleaves or conifers based on the dominant tree types at each site (≥50% cover). Fungal (**b, d**) and bacterial (**c, e**) community compositions in the organic (**b**; $n = 209$; **c**; $n = 255$) and mineral (**d**; $n = 195$; **e**; $n = 266$) soil horizons. Sample size variations result from samples unable to be amplified for fungal and/or bacterial profiling or those not meeting quality control standards of sequencing depth. Vectors show correlations with forest abiotic and biotic variables with principal coordinate analysis (PCoA) axes 1 and 2 where each environmental variable is predicted by PCoA axes 1 and 2 using multiple regression. Solid lines with black labels show significant correlations ($P \le 0.05$) while dashed arrows with gray text show non-significant correlations. The significance of fitted vectors and factors was tested using 999 permutations of environmental variables. Squared correlation coefficients for each variable with respect to PCoA axes 1 and 2 for fungi (panel with left-orientation bars) and bacteria (panel with right-orientation bars) (**f**). *N dep.* nitrogen deposition, *MAT* mean annual temperature, *MAP* mean annual precipitation, *C stocks* soil carbon stocks.

linked to microbial composition, which highlights that there is unique variation in soil microbiomes that could contribute to variation in carbon storage outcomes that cannot be explained by the environment alone. We therefore built and compared statistical models to explore which dimensions of microbiome biodiversity are most strongly linked to forest tree growth and biomass carbon stocks after accounting for other important co-variables.

We demonstrate that both fungal community composition (principal coordinate analysis axis 1; PCoA1; Fig. 2a, e), and now fungal richness (Fig. 2c, g) are correlated with tree biomass and rates of tree growth, even after statistically controlling and accounting for the influence of other important environmental co-variables (Supplementary Data 1, Supplementary Table 1). Tree growth was more strongly linked to fungal composition compared to fungal richness, indicating that which species are present could have larger impacts on tree growth than the overall number of species in a community. These links were also stronger in conifer versus broadleaf forests, but comparable correlations were observed in both stand types (Supplementary Fig. 2). Tree growth was not correlated with bacterial composition (Fig. 1b, f) nor richness (Fig. 1d, h). Tree growth was also more strongly correlated with fungal composition in the mineral versus organic horizon (Supplementary Data 1 and Supplementary Table 2), potentially because most tree roots grow in the mineral horizon[61] and symbiotic, ectomycorrhizal fungal relative abundances were higher in

mineral compared to organic horizon soils (Supplementary Table 3). To that end, the composition of ectomycorrhizal fungi followed by endophytes was most tightly linked to tree growth rates (Fig. 3a). This is consistent with an earlier root tip survey of ectomycorrhizal fungal communities[7] based on an entirely independent sampling effort and medium (individual root-tip sequencing rather than whole soil DNA sequencing). We also discovered that fungal endophyte richness was strongly and positively linked to tree growth rates (Fig. 3a, b), followed by richness of saprotrophs, wood-decomposing fungi (a subgroup within the saprotroph community), plant pathogens, and ericoid but not ectomycorrhizal fungi. The endophyte richness effect size was approximately one third higher than all other groups, even when accounting for other co-variables (Supplementary Table 4). Unlike other fungal guilds, endophytic and ectomycorrhizal fungi are both mutualistic, tree-biotrophic groups in these forests, which might explain why both groups were more connected to forest tree growth compared to other fungi.

All plants in the environment associate with fungal endophytes that can profoundly impact plant fitness. Yet, the ecological significance of fungal endophytes in forests is surprisingly understudied compared to pathogens and mycorrhizae. Most endophyte research is conducted in grasslands[62,63], but it is possible that some of these findings can be generalized to forest systems. Endophytes can promote plant growth via phytohormone production, protection against

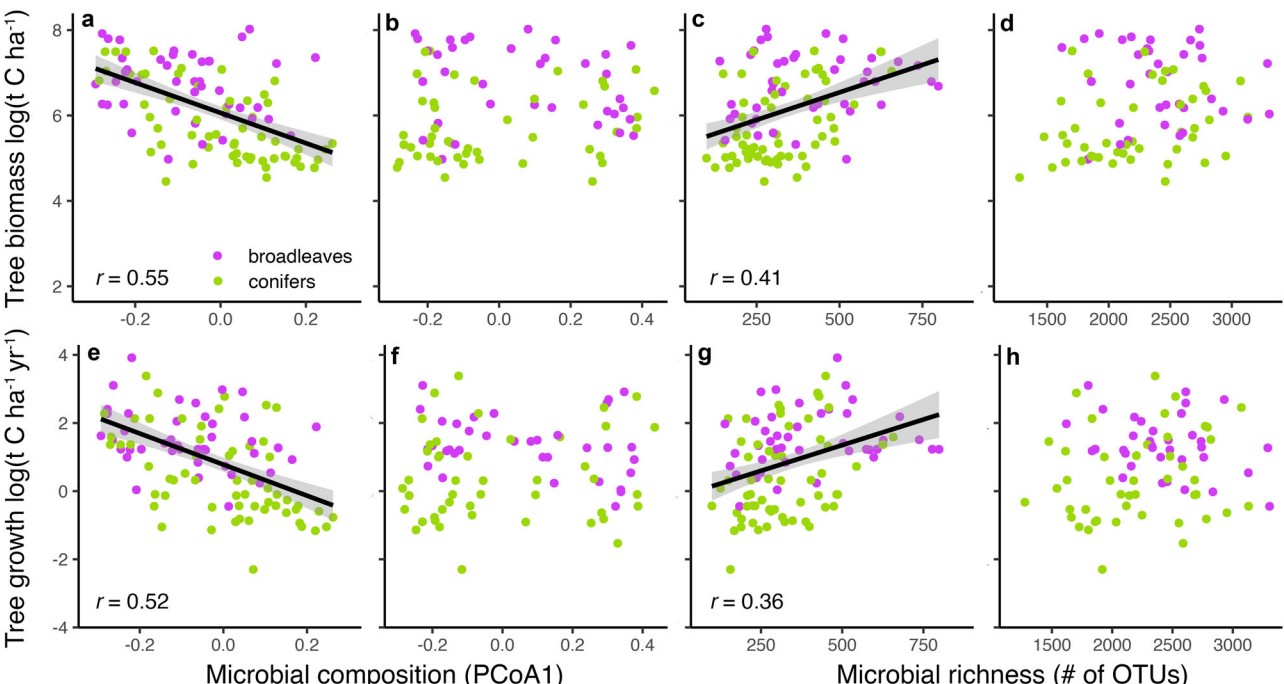

**Fig. 2 | Correlations between microbiome community composition and richness and forest tree growth and biomass.** Panels showing that total fungal composition (principal coordinate analysis axis 1; PCoA1; **a**, **e**; $n = 112$) and richness (**c**, **g**; $n = 112$), but not bacterial composition (**b**, **f**) and richness (**d**, **h**), are correlated with tree biomass and tree growth rates. Plotted lines show linear correlations, shaded areas around each line are 95% confidence intervals, and $r$ values are Pearson correlation coefficients. Communities from the mineral soil are shown here because they were more tightly correlated than organic horizon communities. The full statistical models, including all co-variates, for each correlation are shown in Supplementary Data 1, Supplementary Table 1. Only significant correlation coefficients are shown ($P \leq 0.05$).

pathogens, increased nutrient uptake, and abiotic stress alleviation[64]. Nevertheless, the benefits provided by endophytes to plants might not always be reciprocal with plant investments, especially for some groups such as dark-septate endophytes which were common in our study system[65]. However, it is particularly interesting that the endophytic indicator species we identified in our study (Fig. 3c, d) include *Trichoderma citrinoviridie* and *T. koningii* which produce high levels of the plant growth promoting hormone, indole acetic acid, compared to other fungi with known endophytic life cycles[66], and their relative abundances were positively correlated with broadleaf tree growth in our study (Supplementary Data 2). We also identified four putatively endophytic *Mortierella* indicators of fast conifer and broadleaf tree growth. In agricultural systems, many *Mortierella* stimulate indole acetic acid production, reduce abiotic stress levels, and improve access to phosphorus and iron[67], with similar positive effects recently observed in tree seedlings[68]. This highlights these taxa as important for future research in forestry applications. Our results suggest that forest fungal endophyte richness and species identity may be key components of forest biodiversity-ecosystem function relationships.

It is important to note that most endophytic fungi, including *Mortierella*, can live saprotrophically, which is one reason we detect them in soil samples where roots were removed. However, much like ectomycorrhizal fungi, some dark-septate endophytes also grow hundreds of meters of extraradical hyphae per gram of soil[69], including endophytes present in our soil samples. Taxa annotated as endophytes in our study most likely have mixed ecological strategies and were detected in both biotrophic and saprotrophic states, a limitation of our study since we cannot identify the precise trophic strategy employed by fungi with endophytic capacities in our samples. This is why we did not separate endophytes into "pure" and "mixed" ecological groups, as we did for ectomycorrhizal fungi where there is clearer albeit still ambiguous trophic division. However, the distinction between soil and roots is constantly obscured as roots modify nearby soil, giving rise to

conditions where similar endophyte communities may reside inside roots and the surrounding soil[70]. This suggests that even though some of the endophytic fungi we observed were probably living saprotrophically, they are still indicators of taxa that form symbioses with trees.

While every significant endophyte indicator was positively correlated with tree growth, the other ubiquitous biotrophic group – ectomycorrhizal fungi – included species linked to both slow and fast tree growth (Fig. 3c, d; Supplementary DatV). This is consistent with our earlier root tip survey of ectomycorrhizal fungi[7]. We identified numerous *Russula* and Cortinariaceae (including the entire genus *Cortinarius*) taxa significantly linked to variation in tree growth. In general, the most indicative species of fast tree growth were *Russula* species (Fig. 3c) whereas *Cortinarius* and *Inocybe* indicator species were the topmost negatively correlated OTUs with tree growth. *Cortinarius* are energy demanding species that produce extensive biomass[71], fungal rhizomorphs[72], and extracellular enzymes[73]. In contrast to some *Russula*, both *Cortinarius* and *Inocybe* also actively assimilate nitrogen from organic sources as deep as 30 cm belowground[74], which not only requires producing exploratory mycelium to vertically tunnel deeper into denser soil but also requires costly oxidases and proteases to access organically bound nitrogen. Because mycorrhizal fungi obtain their energy from host-trees, these "costly" traits may constrain tree growth compared to certain, less energy-demanding *Russula* species. *Russula* itself is a large genus containing nitrophobic and nitrophilic species[75], which could explain why we detected *Russula* indicators of both slow and fast tree growth rates for both conifer and broadleaf forests in our study. Importantly, local environmental conditions such as forest succession[76], soil pH[77], drought[78], and nitrogen deposition[79] also select for these particular ectomycorrhizal taxa, which in turn shapes their distributions and potential impacts on forest tree growth. Thus, any potential effects of these fungi on tree growth are likely modulated by environmental

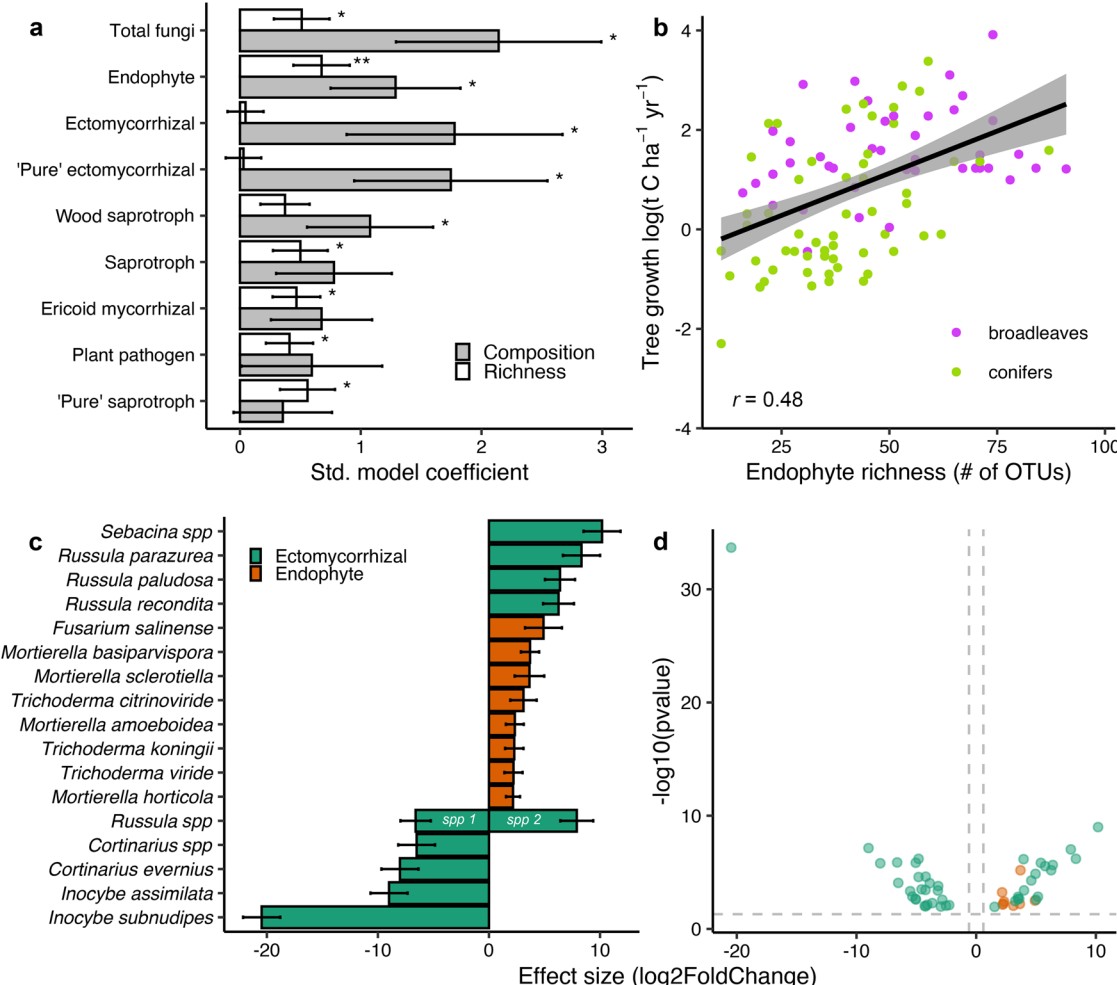

**Fig. 3 | Fungal functional and taxonomic components of species richness and composition significantly linked to tree growth.** Generalized additive model coefficients showing the slope of the linear relationship between tree growth and fungal richness or fungal composition that is independent of all other co-variables (**a**). The standardized model coefficient (bars) and standard error (error bars) are shown so each predictor is on the same scale but note that the effect size of richness and composition are on different scales and should not be directly compared. Separate models were made for each fungal functional group. 'Pure' refers to analyses with fungi only identified to one versus multiple functional categories and asterisks (*) indicates a significant effect (*$P \leq 0.05$, **$P < 0.01$; see Supplementary Table 4 for full statistical model summaries, including exact $P$-values). Correlation between endophyte fungal richness and tree growth rate (**b**; $n = 112$). The plotted line shows the linear correlation, shaded area around the line shows the 95% confidence interval, and the $r$ value is the Pearson correlation coefficient. The top biotrophic fungal indicator species of variation in tree growth rates (**c**). The top five positively and negatively correlated ectomycorrhizal fungi and all endophytic

fungal indicators are visualized. We only show the top ten ectomycorrhizal fungi because there were too many ectomycorrhizal indicator species to fit in one graphic (see Supplementary Data 2 for a complete list). And we only show ectomycorrhizal fungi and endophytes because both are the major biotrophic fungal group in European forest soils, and their compositions both predicted tree growth rates. Bars with a genus level designation are OTUs that could not be identified at the species level. For example, there are two *Russula* OTUs with opposite directional effect sizes distinguished as species (*spp*) 1 and 2. Indicator species were identified as those having significant differential relative abundances based on the negative binomial distribution. Values are reported on a logarithmic scale to base 2 and represent changes in relative abundance for a unit change in tree growth (bars) and their standard error (error bars). Volcano plot showing the strength of all endophytic and ectomycorrhizal fungal OTUs significantly correlated to tree growth (**d**). Values less and more than 0 indicate negative and positive correlations with tree growth, respectively.

conditions. Our results provide support for the idea that ectomycorrhizal symbiosis spans a large spectrum of outcomes for plant growth in forest ecosystems[80].

**Which microbiome metrics predict soil organic carbon stocks?**
Fungal (Fig. 4a) and bacterial (Fig. 4c) community compositions were correlated with organic horizon carbon stocks, though this correlation was more than two times stronger for bacteria compared to fungi, even after accounting for other co-variables of known importance (see Supplementary Data 3), including soil clay content[81], climate[80], forest type[82], and nitrogen deposition[83]. Like microbiome composition, both fungal (Fig. 4b) and bacterial (Fig. 4d) richness were also negatively correlated with organic horizon carbon stocks, and this effect was

relatively larger for bacteria compared to fungi and less driven by dominant tree type differences (i.e., broadleaf forests possessing higher soil microbial diversity and lower organic horizon carbon stocks than coniferous forests). Like tree growth, organic horizon carbon stocks were more tightly linked to microbiome composition than richness. Bacterial, but not fungal, composition was also correlated with mineral horizon carbon stocks ($P = 0.006$), but this effect size was approximately one sixth of that observed in the organic horizon (Supplementary Fig. 3). Neither bacterial richness nor fungal composition/richness were correlated with mineral horizon carbon stocks. This shows that while fungal composition is correlated with mineral horizon carbon stocks alone (Fig. 1f), this correlation is not robust when we account for other co-variables. Thus, in contrast to

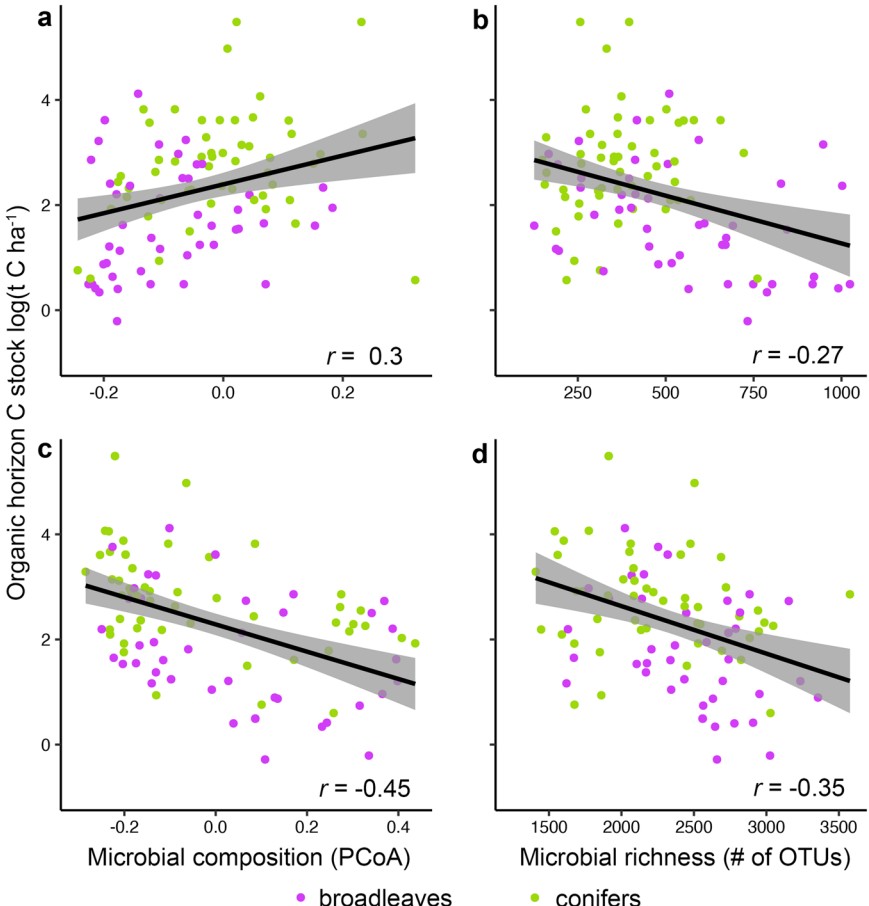

**Fig. 4 | Correlations between carbon (C) stocks and the soil microbiome in the organic horizon.** Panels showing that total fungal composition (principal coordinate analysis axis 2; PCoA2; **a**; $n = 106$) and richness (**b**; $n = 106$) as well as bacterial composition (PCoA1; **c**; $n = 104$) and richness (**d**; $n = 104$) are correlated with tree growth rates. Because there were no correlations in the mineral horizon (except for a weak correlation between bacterial composition and soil organic carbon stocks, see Supplementary Fig. 3), communities and carbon stocks from the organic horizon alone are shown. Plotted lines show linear correlations, shaded areas around each line are 95% confidence intervals, and $r$ values are Pearson correlation coefficients. The full statistical models, including all co-variates, for each correlation are shown in Supplementary Data 3. Only significant correlation coefficients are shown ($P \leq 0.05$).

tree growth, bacterial communities were more tightly correlated with carbon stocks in the organic horizon compared to fungal communities, and bacteria were the only group correlated with mineral horizon carbon stocks.

The relationship between biodiversity and ecosystem functioning is of long-standing ecological and conservation interest. An open question is why fungal and bacterial richness is negatively correlated with organic horizon carbon stocks in our study system. On one hand, this is surprising because the species-energy hypothesis predicts that increasing carbon inputs to a system should boost biodiversity[84], and thus, we should expect microbial richness to positively co-vary with carbon stocks. We would also expect this based on the species-area relationship[85], where less organic horizon carbon means less habitat for species co-existence. In contrast, the biodiversity-ecosystem function concept challenges the idea that environmental conditions alone determine species diversity[86] and argues that higher levels of diversity increase rates of emergent biological processes such as productivity and decomposition. A major soil carbon loss pathway is decomposition, which can be enhanced by microbial richness in European forests under certain scenarios[87]. A recent meta-analysis also shows that experimental reduction of bacterial and fungal diversity decreases soil respiration[10], and microbial richness is positively correlated with decomposition rate in multiple observational studies[23,29]. We will not disentangle the directionality or causality of these correlations in our study, we can raise this as a subject for future

investigation. A notable starting point would be to explore the links between microbial richness and soil carbon storage while experimentally removing confounding effects of soil pH. Microbial richness is often linked to soil pH[46,49], and soil pH is correlated with soil carbon stocks in our study (Supplementary Data 4). While soil pH was not so strongly correlated with microbial richness in our study system, and the statistical effects of richness and soil pH on soil carbon stocks were independent, we cannot disentangle the possibility that microbial richness is largely structured by soil pH and thus, only indirectly linked to soil carbon stocks.

While the link between microbial richness and carbon storage is particularly interesting in the context of conflicting ecological theories, it is important to emphasize that bacterial composition was more strongly linked to carbon stocks than species richness. We therefore explored which bacterial lineages were positively and negatively linked to organic horizon carbon stocks. At the OTU-level, there were both positive and negative indicators of organic horizon carbon stocks within most major phyla. Eighty percent of the Proteobacteria [Pseudomonadota] indicator OTUs were negatively correlated with carbon stocks in conifer forests (Supplementary Fig. 4). Relative abundances of Proteobacteria in conifer stands was also positively correlated with soil pH ($r = 0.24$, $P = 0.02$; Supplementary Fig. 5), which is negatively correlated with organic horizon carbon stocks independent of the microbiome (Supplementary Data 4). An indirect tie to soil pH could further explain this lineage's link to organic horizon carbon

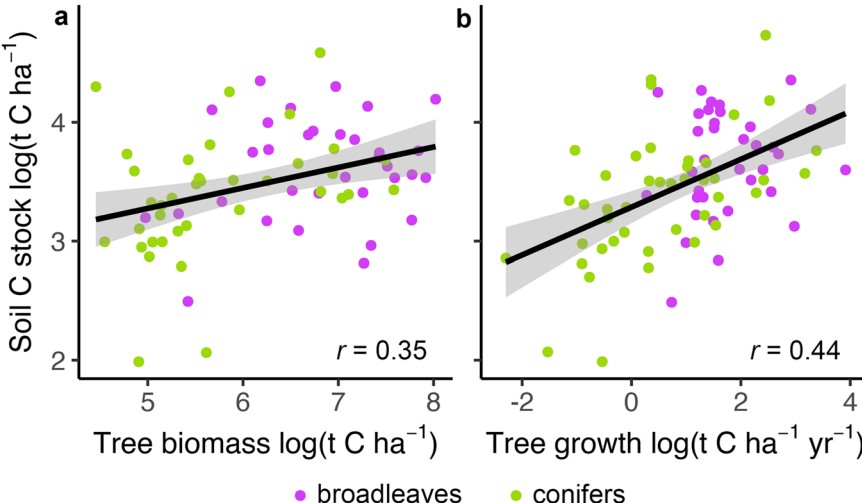

**Fig. 5 | Correlation between tree growth and biomass and soil organic carbon (C) stocks.** Panels show the correlation between tree biomass (**a**) and growth (**b**) with soil organic carbon stocks in the mineral horizon (0–10 cm depth; $n = 88$). Plotted lines show linear correlations, shaded areas around each line are 95% confidence intervals, and $r$ values are Pearson correlation coefficients. The full statistical models, including all co-variates, for each correlation are shown in Supplementary Data 4. Only significant correlation coefficients are shown ($P \leq 0.05$).

stocks. For fungi, we identified two ectomycorrhizal fungal Russulaceae OTUs (*Lactifluus vellereus* and *Russula rhodopus*) positively linked to organic horizon carbon stocks in conifer stands (Supplementary Fig. 6). These ectomycorrhizal fungi might slow decomposition via the Gadgill effect[88], an expected outcome of some ectomycorrhizal fungi in conifer forests[89]. Though other ectomycorrhizal lineages with stronger decomposing potentials, such as *Cortinarius* or *Piloderma*, would be the more anticipated fungi responsible for the Gadgill effect, ectomycorrhizal genera with weaker decomposing potentials have been recently linked to slower carbon cycling[90]. Many more fungi were indicators of organic horizon carbon stocks in broadleaf versus conifer forests (120 versus 11 OTUs). Most top positive indicators were ectomycorrhizal fungi, whereas the top negative indicators were mostly saprotrophs, with or lacking mixed trophic assignments (e.g., saprotroph-pathogen), implying that ectomycorrhizal fungal indicators are more strongly tied to higher organic horizon carbon stocks. Among the top indicators were various *Inocybe*, *Sebacina*, and *Russula* OTUs (Supplementary Data 5), all ectomycorrhizal fungi that we also found to be strongly linked to variation in tree growth (Fig. 2c). These OTUs, many of which we classified at the species-level so they can be investigated more directly, are unique since they are indicators of carbon cycling both above- and belowground.

### Conclusion and limitations

In this study, we linked soil microbiome composition and diversity to three major forest carbon storage metrics across Europe (tree growth, tree biomass, and soil organic carbon stocks). We show that fungal, but not bacterial, composition and richness are correlated with tree growth rates and tree biomass carbon stocks, when controlling for the effects of climate, dominant tree type, and other important co-variables. We suspect a major reason for opposing fungal and bacterial signals aboveground is the ubiquity of biotrophic fungal groups in the forest mycobiome and key groups of symbiotic endophytic and ectomycorrhizal fungi. Fungal endophyte biodiversity was positively linked to tree growth rates above and beyond any other microbial group we studied, a surprising discovery since most work in forests highlights mycorrhizal fungi[7,12,30,31,91,92]. It is important to note that there are also many biotrophic bacteria in forests, including endophytes[93], but these are especially difficult to identify based on DNA sequencing alone and could not be confidently separated from the entire bacterial community in our study. Future work will need to

further investigate our fungal endophyte-tree growth results to explore any causal relationship between the two. That said, the biotechnology sector has already begun widely capitalizing on fungal endophytes for applied tree growth promotion[94]. By sourcing specific endophytes or communities from forested areas in the wild, locatable using observational studies like ours, it would be possible to test whether even more powerful plant biostimulants can be developed.

Even though tree growth is a major component of forest carbon cycling, an equal or even greater quantity of carbon is stored belowground[2]. Both fungal and bacterial composition and richness were negatively correlated with organic horizon carbon stocks, implying a potential direct link between the two that will need to be disentangled further with experimental studies to investigate any cause-effect interactions. In the mineral soil horizon, where most carbon is stored, the microbiome was not tightly linked to soil organic carbon stocks. However, tree growth and biomass were tightly linked to mineral horizon organic carbon stocks in our study system (Fig. 5) even after controlling for other co-variables (Supplementary Data 4). Since the mycobiome was tightly correlated with tree growth/tree biomass, it is therefore indirectly linked to mineral horizon organic carbon stocks and in turn total ecosystem carbon storage. This establishes that the soil mycobiome is a unique biological indicator of forest carbon storage across Europe.

## Methods

### Study sites

This work was conducted across the ICP Forests network which has been monitoring hundreds of permanent forest plots across Europe since the 1990's[95]. We sampled level II plots that are intensively monitored, at least 0.25 ha, and where almost all trees with a > 5 cm diameter at breast height (DBH) are measured approximately every five years, a common interval for estimating tree growth. At each plot, we measured tree species membership and whether the plot was dominated by conifer versus broadleaf trees (>50% cover). There were 21 tree species included in the survey, and tree richness ranged from 1 to 9 tree species. Most plots were between 1 and 5 tree species. Forest age ranged from <30 years old to >120 years old with an average age of 90 years. The locations spanned a -2.5 to 15.5 °C mean annual temperature range, a 443 to 2,082 mm year$^{-1}$ mean annual precipitation range, and a 0.10 to 50.11 t C ha$^{-1}$ year$^{-1}$ productivity range.

## Soil sampling

Soil was sampled between July-August in 2019 and 2020 from 285 level II plots across 18 European countries. We were only able to include 238 plots from 15 countries in our study due to issues extracting DNA and amplifying microbial marker genes. A 30 ×30 m subplot was established inside the plot or in the buffer zone, and nine samples were collected in a grid-design (Supplementary Fig. 7). The organic horizon was first removed using a serrated knife and spatula (only at sites where it was formed and separable), and mineral soil was collected to a 10 cm depth using a soil corer (5 cm diameter). Soil samples were pooled within horizon, homogenized, and dried in an oven at 40 °C or air-dried for at least 48 hours, depending on whether a drying oven was available. Fully dried samples were then shipped to ETH Zürich and stored at -20 °C prior to analysis.

## Carbon cycling and meta-data measurements

Each level II monitoring plot collects in situ tree, vegetation, soil, climate, and atmospheric chemistry data. Tree growth was calculated using periodic DBH measurements and allometric equations for each tree species and DBH size range. In short, we removed any dead trees, trees with <5 cm DBH, trees that shrank over the growth period, and then used the first and last DBH measurement to calculate diameter growth increment. The mean growth interval was 5.5 years, the mean initial year was 2005, and the mean final year was 2008. While this varies marginally from the time of soil sampling, previous work has demonstrated that year-to-year variation in microbiome composition is rather low[96,97]. Next, we used species specific allometric equations from publications made studying trees in Europe within the size range of those observed in our dataset to compute tree mass at the first and final census (see[7]), computed tree growth mass, and assumed a 50% C content across all species[98]. Because every tree in a plot is not measured for DBH, we could not strictly sum the mass of all measured trees to go from the tree to stand level. We therefore randomly sampled with replacement trees which are periodically measured until reaching in situ stem density 1,000 times and used the mean value to estimate stand-level tree growth rates (tonnes C ha$^{-1}$ yr$^{-1}$) and live tree biomass (tonnes C ha$^{-1}$).

Soil carbon and nitrogen stocks were calculated using measurements of elemental content (%), bulk density, and sampling depth (tonnes C ha$^{-1}$) determined from field-based measurements. Soil carbon and nitrogen contents were measured using dry combustion on finely ground soil samples. Soil pH was measured in soil slurries with DI water (10 g soil: 20 mL DI water) using a pH probe. Soil clay content was measured in situ and also estimated using SoilGrids at a 250 m resolution[99]. However, this data-product-derived estimate was only used after we assessed the accuracy of these estimates using in situ data. We compared estimated values to those collected in the lab for a subset of the plots where data was available across the entire ICP level II network. The two values were strongly correlated ($r = 0.51$, $P < 0.0001$, $n = 321$), so we used data from SoilGrids to have more complete observations except when modeling soil carbon storage. For soil carbon storage, we used in situ clay data due to the importance of clay in stabilizing soil carbon. Finally, we obtained mean annual temperature and precipitation measurements from WorldClim[100], and N deposition predictions for 2019 at a 1 km resolution from EMEP[101].

## Molecular analyses

DNA was extracted from frozen soil (250 mg) using the DNeasy PowerSoil Pro kit (Qiagen, Hilden, Germany). Template DNA was then used to amplify the variable regions 4 and 5 of the 16S rRNA gene using the primers 515F (GTGYCAGCMGCCGCGGTAA) + 926R (GGCCGYCAATT YMTTTRAGTTT)[102] to study prokaryotes and the entire ITS region using the primers ITS9munngs (GTACAC ACCGCCCGTCG)+ ITS4ng (CGCCTSCSCTTANTDATATGC)[103] to study fungi. The 16S primers

were selected because they offer improved phylogenetic resolution compared to the use of alternative reverse primers[104], and the ITS primers were selected because they also span a wide phylogenetic range of fungi and best recapitulate mock communities compared to other primer combinations[103]. Each primer contained a 12 bp index sequence in the 5' position. PCR reactions were performed in duplicate 25 μL reactions (13 μL of PCR grade water, 10 μL of Phusion Flash High-Fidelity PCR Master Mix, 1 μL 12.5 μM forward primer, 1 μL 12.5 μM reverse primer, and 1 μL of template DNA). 16S amplicon thermocycler conditions were 94 °C for 3 min followed by 30 cycles of 94 °C for 45 s, 50 °C for 60 s, and 72 °C for 90 s, then 72 °C for 10 min, and finally a 4 °C hold. ITS amplicon thermocycler conditions were 95 °C for 15 min followed by 30 cycles of 95 °C for 30 s, 57 °C for 30 s, and 72 °C for 60 s, then 72 °C for 10 min, and finally a 4 °C hold.

The success and relative quantity of PCR product was assessed using agarose gel electrophoresis. We then pooled samples based on band intensity and removed remaining PCR reagents, short DNA and PCR products, and PCR primer dimers using AMPure beads for specific size selection. The ITS amplicons averaged ca. 750 bp whereas the 16S amplicons averaged 300 bp. Pooled products were then quantified on a Qubit using the dsDNA BR Assay Kit (Invitrogen, Waltham, Massachusetts, USA) and sent for library preparation and sequencing at the Functional Genomics Center Zürich. 16S libraries were sequenced using four Illumina MiSeq Runs with v3 chemistry (2 × 300 bp). ITS libraries were sequenced using four PacBio Sequel IIe SMRT Cell 8 M (15 h movie lengths).

## Bioinformatics

Raw sequences were first demultiplexed using Cutadapt[105] allowing for 0.10% mismatch, no insertions or deletions, and using the –pair-adapters function. 16S reads included (forward) F and (reverse) R reads whereas ITS sequences were HiFi reads produced using the circular consensus sequencing mode. The accuracy of HiFi reads provides a base-level resolution of 99.9% accuracy. Demultiplexed sequences were then imported into QIIME2 (v2021.8) for downstream processing[106]. However, prior to importing the ITS sequences, we first extracted the complete ITS region using ITSx (v1.1.3)[107]. 16SF and R reads were first merged using the vsearch join-pairs plug-in derived from USEARCH[108]. Because the ITS reads are single end, there was no need for pairing. We then QC filtered all reads using the quality-filter q-score command removing reads with average PHRED scores <4, truncating reads if >3 successive base call PHRED scores were <3, and removing all sequences with ambiguous base calls. We then dereplicated sequences and clustered de novo operational taxonomic units (OTUs) at 97% sequence similarity for 16S sequences and 98% sequence similarity for ITS sequences to account for variation in sequence conservation and better capture species identities compared to computing amplicon sequence variants or ASVs[109] using the dereplicate-sequences and cluster-features-de-novo functions, respectively. Previous research also shows that computing ASVs or OTUs makes little difference when detecting patterns in community composition, species richness, and relative abundances of taxa for both 16 S and ITS DNA metabarcoding[110]. Singletons were later removed from the dataset in R. Finally, we assigned taxonomy to representative 16 S and ITS OTUs using Greengenes[111] (2019-05 release) and UNITE[112] (v8, 2021-10 release), respectively. We used the naïve Bayes machine-learning classifier and the feature-classifier fit-classifier-I-bayes function to train the classifier. We then assigned taxonomy using the classify-sklearn function and used the default confidence parameter of 0.7. ITS OTUs were also assigned functional guild annotations at the genus level using FUNGuild[113], accepting all 'probable' or higher level annotations. Where multiple functional annotations were assigned, we grouped them together (e.g., ectomycorrhizal-saprotroph). We also calculated metrics for taxa strictly identified to one

trophic group for ectomycorrhizal fungi and saprotrophic fungi as 'pure' ectomycorrhizal fungi and 'pure' saprotrophic fungi. We did not do this for pathogens or endophytes because these groups are generally expected to harbor >1 trophic strategy. See Supplementary Table 3 for a summary of the relative abundance of different fungal functional group annotations.

## Statistical analyses

**Microbiome diversity and composition.** Samples with low sequencing depth were first removed from the dataset (<5000 sequences for 16S analysis; <500 sequences for ITS analysis). Because PacBio Sequel IIe sequencing is much shallower than Illumina MiSeq (2 × 300 bp) sequencing, we rarified the fungal dataset to a lower depth than the bacterial dataset. Although, sequencing depth in the ITS dataset is relatively low, we find similar correlations with environmental variables when removing low-depth samples and rarefying to 3000 sequences. We therefore opted to retain more samples and rarefed to <500 sequences (see Supplementary Data 6 for raw sequence counts). For estimating alpha and beta diversity, we rarified the datasets to the lowest sequencing depth using the rrarefy function in vegan (2.6-4)[114]. We then calculated relative abundance of OTUs and measured the correlation between microbiome composition and environmental variables used to predict tree growth and soil organic carbon stocks (in addition to latitude referred to as 'geographic space' in Fig. 1) using distance-based redundancy analysis and the capscale function in the vegan package. Analyses were performed separately for bacteria and fungi using Bray−Curtis dissimilarities, and predictor variables were scaled to directly compare effect sizes. We also estimated species richness and Shannon Diversity using the specnumber and diversity functions in vegan, respectively. We then estimated community composition (i.e., beta diversity) based on OTU relative abundances converted to Bray−Curtis dissimilarities to have a metric of community composition for our statistical models of tree growth and SOC stocks. Bray−Curtis dissimilarities were analyzed using principal coordinate analysis (PCoA) and the pcoa function in the ape package (5.6-2)[90] (see Supplementary Table 5 for eigenvalues). We also fit environmental variables to PCoA1 and 2 using the envfit function in the vegan package. Analyses were conducted separately for bacteria and fungi, and then the fungal dataset was split into functional guilds and analyses were repeated separately for each guild. Microbiome alpha and beta diversity (PCoA1 and 2) were then used to predict tree growth, tree biomass, and soil carbon stocks in subsequent regression analyses (see below).

**Indicator species analysis of discrete and continuous variables.** We identified which taxa were linked to continuous variables of C cycling (tree growth rate and soil organic carbon stocks stocks) using analysis of differential relative abundances and negative binomial models. We used the non-rarified OTU table and DESeq2 package (1.34.0)[115]. We used the estimateSizeFactors function with type = 'poscounts' and then the DESeq function with test = 'Wald' and fitType = 'parametric'. Estimated values from this analysis represent a log change in sequence abundance for a unit change in the response variable. Taxa with significant correlations were identified as "indicators species", defined as those with significant $p$-values and log two-fold change >0.6 or <-0.6, consistent with most RNA sequencing workflows. The direction of the model coefficient was used to assess whether they were linked to low or high values of each response variable. Although we use the term "indicator species", this is distinct from those species identified using traditional indicator species analysis[116] which only identifies indicator species linked to discrete groups versus continuous variables.

**Regression analyses.** All statistical analyses were conducted in R[117] and significance was set to $P \leq 0.05$. We used generalized additive modeling (GAMs) to account for the linear and non-linear effects of predictors on tree growth and soil carbon stocks. We used the gam function from the mgcv package (1.8–38)[118] and used REML estimation of the smoothing parameters. We predicted tree growth, tree biomass carbon stock, and soil carbon stocks using nitrogen deposition, soil nitrogen stocks (for tree growth only), mean annual temperature, mean annual precipitation, soil pH, soil clay content, stem density, forest age, and a categorical predictor of broadleaf versus conifer stand type. Each model also contained one microbiome predictor (e.g., PCoA1, PCoA2, species richness) to maintain independence and facilitate model comparisons across different microbiome predictors. We diagnosed model fit based on the distribution of the residuals and confirmed that predictors were not too strongly multi-collinear based on variance inflation factors ≤5[60] (see Supplementary Data 7 for results). To emphasize the correlational nature of our work in the display items, we computed Pearson correlation coefficients ($r$) versus the coefficient of determination from regression analysis ($r^2$).

## Reporting summary

Further information on research design is available in the Nature Portfolio Reporting Summary linked to this article.

## Data availability

Full access to raw ICP Forest datasets is available via the ICP Forests network upon request (http://icp-forests.net/page/data-requests). Restrictions apply to the availability of these data without a formal data request. Raw microbiome datasets can be downloaded from the NCBI SRA using accession numbers PRJNA1068067, PRJNA639984, PRJNA644776, and PRJNA1068308. Microbiome and other data products can be downloaded in the following repository https://gitlab.com/fungalecology/icpf.micro. The fungal taxonomic database UNITE can be accessed here: https://unite.ut.ee/index.php; the bacterial taxonomic database Greengenes can be accessed here: https://greengenes.secondgenome.com/. The fungal functional group database FUNGuild can be accessed here: http://www.funguild.org/.

## Code availability

All scripts used for the analysis are openly available at the following repository: https://gitlab.com/fungalecology/icpf.micro.

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

## Acknowledgements

This work was supported by an Ambizione grant (PZ00P3_179900 awarded to C.A.) from the Swiss National Science Foundation (SNSF). M.A.A. was also partially supported by an Ambizione grant (PZ00P3_208648 awarded to M.A.A.) from the SNSF. The evaluation was based on data (Data Request No 168) that was collected by partners of the official UNECE ICP Forests Network (http://icp-forests.net/contributors). Part of the data was co-financed by the European Commission (Data achieved at "15.02.2022").

## Author contributions

Conceptualization, M.A.A. and C.A.; Formal Analysis, M.A.A.; Investigation, M.A.A., C.A.; Resources, C.A., T.W.C., M.A.A.; Writing—Original Draft, M.A.A.; Review & Editing, M.A.A., L.T., B.D.V, L.C., H.Mee., M.W., H.A., F.J., P.L., A.K., M.G., G.Pop., B.F., A.G., M.S., M.F., P.W., V.C., R.C., G.Pap., A.M., M.I., L.V., P.R., H.Mei., V.T., M.D., N.E., A.S., N.v.T., T.C., C.A.; Supervision, C.A., T.W.C.

## Competing interests

The authors declare no competing interests.

## Additional information

¹Department of Environmental Systems Science, ETH Zürich, Zürich, Switzerland. ²Swiss Federal Institute for Forests, Snow, and the Landscape Research (WSL), Birmensdorf, Switzerland. ³Center for Microbiology and Environmental Systems Science, University of Vienna, Vienna, Austria. ⁴Mycology and Microbiology Center, University of Tartu, Tartu, Estonia. ⁵Environment & Climate Unit, Research Institute for Nature and Forest, Geraardsbergen, Belgium. ⁶French National Forest Office, Fontainebleau, France. ⁷Northwest German Forest Research Institute, Göttingen, Germany. ⁸Sachsenforst State Forest, Pirna OT Graupa, Germany. ⁹Forest Research Institute, Sękocin Stary, Poland. ¹⁰Research Institute for Forest Ecology and Forestry, Trippstadt, Germany. ¹¹Executive Environmental Agency at the Ministry of Environment and Water, Sofia, Bulgaria. ¹²Mediterranean Center for Environmental Studies, Paterna, Spain. ¹³Department of Plant Diversity and Ecosystem Management, University of Camerino, Camerino, Italy. ¹⁴Arma dei Carabinieri Forestry Environmental and Agri-food protection Units, Rome, Italy. ¹⁵Slovenian Forestry Institute, Ljubljana, Slovenia. ¹⁶Department of Geosciences and Natural Resource Management, University of Copenhagen, Frederiksberg C, Denmark. ¹⁷Natural Resources Institute Finland, Rovaniemi, Finland. ¹⁸Division of Forest and Forest Resources, Norwegian Institute of Bioeconomy Research, Ås, Norway. ¹⁹Division of Biotechnology and Plant Health, Norwegian Institute of Bioeconomy Research, Ås, Norway. ²⁰State Agency for Nature, Environment and Consumer Protection of North Rhine-Westphalia, Recklinghausen, Germany. ²¹Thuenen Institut of Forest Ecosystems, 16225 Eberswalde, Germany. ²²Environmetnal Computational Science and Earth Observation Laboratory, EPFL, Lausanne, Switzerland.
✉e-mail: manthony5955@gmail.com

