## [Peer Review File · Nature Communications]

Fungal community composition predicts forest carbon storage at a continental scaleEditorial Note: This manuscript has been previously reviewed at another journal that is not operating a transparent peer review scheme. This document only contains reviewer comments and rebuttal letters for versions considered at *Nature Communications*.

Reviewer #2 (Remarks to the Author):

I am satisfied with the revision! Thank you for this amazing work!

Reviewer #3 (Remarks to the Author):

Unique signature of fungal biodiversity in continental-scale forest carbon storage

The authors have substantially improved the manuscript based on previous revisions. Most importantly, the concern of causality has been improved, and the results are more tempered with a more thorough discussion. There are still some issues with the presentation of the results and the interpretation given the usage of multiple tree hosts, and FunGuild.

Figure 1 – what is 1f showing? Is one side bacterial and one is fungal? Needs more clear label or description in the caption.

Figure 1-2, it would be helpful to add headings of which are fungal and bacterial plots to further assist the reader. If you don't feel this is stylistically necessary, I also understand

Line 107: please rephrase decompose to 'decay'. Sense Lindahl and Tunlid 2015 *New Phyt* and Zak et al., 2019 *New Phyt*.

Results and discussion:

Please report the axis weightings for the ordinations, so that the relative importance of PCOA1 can be understood.

Line 162: What is the underlying data supporting this statement? If it is Figure 1b/d, I cannot understand how this statement is supported.

One big concern I have corresponds to the fact that fungal communities in broadleaf and coniferous forests are so strongly differentiated. Accordingly, host effects strongly influence the findings here. This is particularly notable in the PCOA Figure 1b/d (i.e. the clustering of the purple/green points), and seems to therefore propagate throughout the ms. It seems that this has been taken into account in the GAM fits, but it might be worth a line or two of discussion on this point, given that it also propagates in Figure 4, quite strongly. To this point, the relative importance of tree host is a stronger predictor than PCOA1 in the GAM fits, this is perhaps not surprising, and should be emphasized in the manuscript. Accordingly, the interpretation should be made that it is not the fungal communities influencing tree growth per se, but potential variation in the reliance of these communities based on tree host identity that seems to be driving this result. This is a finer point, and accordingly, (with the below comment), I think helps to get at some interesting biology.

One area where this manuscript would be much improved is analyzing the effect of soil communities on Broadleaf and Conifer trees independently. Because the importance of tree host is so important in the GAM fits, it would be interesting to directly compare the relative effect of microbial communities on these hosts independently. This would add some necessary nuance to the manuscript and help clarify interactions among different tree lineages and their reliance on fungal/bac communities.

Line 195, could this point be made in a figure, or a supplementary table? I presume this is a GAM output. This paragraph is a positive step forward, however, I am still needing further clarification. Can line 197 be rewritten in such a way as to improve this point?

In the comments to previous reviews, you made the following statement:

"We used FUNGuild for this analysis. Rather than calling a fungus a "wood saprotroph", as it appears in the raw FUNGuild output, we annotated these fungi as "white rot" since there was no wood in our soil samples. We now include this information in the Table caption for Table S4 where we write: 'Italicized groups are those within the group above (e.g., white rot fungi are a group within the saprotrophs annotated as "wood saprotrophs" in FUNGuild).'"

However, a 'wood saprotroph' is not necessarily a 'white-rot' saprotroph, and this point needs to be clarified (e.g., brown rot fungi are also considered wood saprotrophs). I recognize that there is minimal wood in soil, and that this is a minor point, but perhaps just calling them 'wood saprotrophs', and making the point that this is a limitation of funguild, is a better strategy than falsely assigning them to a specific life-style and enzymatic repertoire.

The added paragraph clarifying that endophytic fungi can also have saprotrophic lifestyles was useful (starting line 262). However, attributing soil fungi with a presently endophytic stage as influencing plant growth is tenuous and should be highlighted as a limitation of this study. For instance, there is a real potential for fungi that are in the soil compartment to be falsely highlighted as influencing plant growth in a spurious manner. Please describe this limitation in the section that you have added.

The paragraph linking observed effect sizes of different species with the life histories of genera provided valuable contextualization of findings (starting line 300).

Line 403 misspelled "Gadgill"

Throughout the results and discussion section, please highlight the relative importance of the non-microbial model predictors (e.g., the results in Table S7). While this paper is focused on the impacts of the microbial community, it is important to also note that other model predictors like soil pH, MAP, and tree type were significant in most models.

Conclusions:

Line 419 misspelled "carbo"

Supplement:

I am missing in the Supplement the number of sequences recovered across sites. Were there vastly different recovery of sequences across plots?

REVIEWER COMMENTS

Reviewer #2 (Remarks to the Author):

I am satisfied with the revision! Thank you for this amazing work!

Thank you very much for your feedback which improved our manuscript a lot!

Reviewer #3 (Remarks to the Author):

Unique signature of fungal biodiversity in continental-scale forest carbon storage

The authors have substantially improved the manuscript based on previous revisions. Most importantly, the concern of causality has been improved, and the results are more tempered with a more thorough discussion. There are still some issues with the presentation of the results and the interpretation given the usage of multiple tree hosts, and FunGuild.

Thank you for the constructive feedback. We have now carefully revised the text to address each of your outstanding comments. Each comment and the revision we made is outlined below.

Figure 1 – what is 1f showing? Is one side bacterial and one is fungal? Needs more clear label or description in the caption.

We have now added additional details to the figure caption to highlight which panels reflect fungal versus bacterial results. In the figure caption, we now write:

“Correlation coefficients for each variable and PCoA axes 1 and 2 for fungi (panel with left-orientation bars) and bacteria (panel with right-orientation bars) (f).”

Figure 1-2, it would be helpful to add headings of which are fungal and bacterial plots to further assist the reader. If you don't feel this is stylistically necessary, I also understand

We have tried this, and it disrupts the aesthetics quite extensively because the words appear too many times in the individual panels of a given figure.

Line 107: please rephrase decompose to 'decay'. Sense Lindahl and Tunlid 2015 New Phyt and Zak et al., 2019 New Phyt.

Done!

Results and discussion:

Please report the axis weightings for the ordinations, so that the relative importance of PCOA1 can be understood.

D
o
n
e

—

Line 162: What is the underlying data supporting this statement? If it is Figure 1b/d, I cannot understand how this statement is supported.

This statement is supported by Figure 1b-f because the correlations between fungal composition and tree growth are only significant for fungal compared to bacterial communities. We have now clarified that this is unique relative to bacteria in the text by revising this last sentence in the paragraph. It now says at lines 164-165:

“Fungal, not bacterial, composition is therefore a uniquely informative marker of forest productivity in addition to forest type and age.”

One big concern I have corresponds to the fact that fungal communities in broadleaf and coniferous forests are so strongly differentiated. Accordingly, host effects strongly influence the findings here. This is particularly notable in the PCOA Figure 1b/d (i.e. the clustering of the purple/green points), and seems to therefore propagate throughout the ms. It seems that this has been taken into account in the GAM fits, but it might be worth a line or two of discussion on this point, given that it also propagates in Figure 4, quite strongly. To this point, the relative importance of tree host is a stronger predictor than PCOA1 in the GAM fits, this is perhaps not surprising, and should be emphasized in the manuscript. Accordingly, the interpretation should be made that it is not the fungal communities influencing tree growth per se, but potential variation in the reliance of these communities based on tree host identity that seems to be driving this result. This is a finer point, and accordingly, (with the below comment), I think helps to get at some interesting biology.

Thank you for this comment. There are a few points to consider here, but the most important is that while fungal communities differ between broadleaf and conifer forests, tree type does not explain the continuous variation observed within a particular forest type nor how this continuous variation is linked to tree growth. We observed a high degree of spatial dissimilarity in fungal composition within a particular tree type (Figure 1). One might incorrectly argue that we are just seeing significant regression results because communities happen to sort in a continuous manner with tree growth simply because broadleaves accrue more carbon when they grow compared to conifers, but this is inaccurate. When we split the dataset by tree type, the alpha and beta diversity of fungal communities is still correlated with tree growth in both stand types (see the next point for elaboration on this). There is also a considerable amount of overlap in fungal composition in stands with broadleaves and conifers, which explains why tree type only partially structures fungal communities. Your assessment therefore reflects one possible scenario that might explain some differences between broadleaf and needleleaf forests, which we talk about already in the manuscript, but it does not consider continuous variation within these groups in a given forest type. Any potential direct and indirect effects of fungal communities on tree growth is not strictly the influence of host plant filtering, even though it certainly does strongly impact fungal composition (*sensu* Figure 1). This is why we explored differences within host tree types in the manuscript, for example, in our indicator species analysis where we conducted all finer-resolution analyses separately by tree types, and now, with additional analyses in our revision (as described below in the proceeding comment).

One area where this manuscript would be much improved is analyzing the effect of soil communities on Broadleaf and Conifer trees independently. Because the importance of tree host is so important in the GAM fits, it would be interesting to directly compare the relative effect of microbial communities on these hosts independently. This would add some necessary nuance to the manuscript and help clarify interactions among different tree lineages and their reliance on fungal/bac communities.

Thank you for this suggestion. We already conducted finer-scale, microbial analyses separately by tree type to dissect general trends observed in our GAM analyses (e.g., indicator species analysis). This already provides more nuance around which aspects of the mycobiome drive observed patterns at the community level in the GAMs. However, we have now additionally examined and visualized different correlations between fungal composition and fungal richness with respect to tree growth. This effort shows that the magnitude of these correlations is stronger for conifers compared to broadleaves, but the same correlations are nevertheless observed in both forest types. We now report this at lines 216-218 and show the correlations split by tree type in the supplement. The text now says:

“Tree growth was more strongly linked to fungal composition compared to fungal richness, indicating that which species are present could have larger impacts on tree growth than the overall number of species in a community. These links were also stronger in conifer versus broadleaf forests, but comparable correlations were observed in both stand types (Figure S2).”

Line 195, could this point be made in a figure, or a supplementary table? I presume this is a GAM output. This paragraph is a positive step forward, however, I am still needing further clarification.

The importance of other, non-microbial predictors of tree growth is acknowledged by citing earlier research. However, the variance inflation factors can be reported upon from our study. We have now added this as a supplementary Table (See **Table S12**). At line 664, we now write:

“We diagnosed model fit based on the distribution of the residuals and confirmed that predictors were not too strongly multi-collinear based on variance inflation factors $\leq 5^{60}$ (see Table S12 for results).”

Can line 197 be rewritten in such a way as to improve this point?

This sentence has been revised to clarify and address a subsequent point you raised. At lines 197-201, it now reads as:

“Here, we show that none of these non-microbial predictor variables are strongly multicollinear with microbiome composition and diversity (variance inflation values ≤ 5 in all models sensu ⁶⁰), but they are important predictors of tree growth in our study (see the supplementary tables referenced throughout the results section).”

In the comments to previous reviews, you made the following statement:

“We used FUNGuild for this analysis. Rather than calling a fungus a “wood saprotroph”, as it appears in the raw FUNGuild output, we annotated these fungi as “white rot” since there was no wood in our soil samples. We now include this information in the Table caption for Table S4 where we write:

‘Italicized groups are those within the group above (e.g., white rot fungi are a group within the saprotrophs annotated as “wood saprotrophs” in FUNGuild).’”

However, a ‘wood saprotroph’ is not necessarily a ‘white-rot’ saprotroph, and this point needs to be clarified (e.g., brown rot fungi are also considered wood saprotrophs). I recognize that there is minimal wood in soil, and that this is a minor point, but perhaps just calling them ‘wood saprotrophs, and making the point that this is a limitation of funguild, is a better strategy then falsely assigning them to a specific life-style and enzymatic repertoire.

Thank you for this fair point. We have now revised all mention to white rot fungi as “wood saprotrophs”. This includes Figure 3 and all tables and figure captions in the supplement.

The added paragraph clarifying that endophytic fungi can also have saprotrophic lifestyles was useful (starting line 262). However, attributing soil fungi with a presently endophytic stage as influencing plant growth is tenuous and should be highlighted as a limitation of this study. For instance, there is a real potential for fungi that are in the soil compartment to be falsely highlighted as influencing plant growth in a spurious manner. Please describe this limitation in the section that you have added.

We have now added an additional sentence to the paragraph to acknowledge this limitation. We now write at lines 274-277:

“Taxa annotated as endophytes in our study most likely have mixed ecological strategies and were detected in both biotrophic and saprotrophic states, a limitation of our study since we cannot identify the precise trophic strategy employed by fungi with endophytic capacities in our samples.”

The paragraph linking observed effect sizes of different species with the life histories of genera provided valuable contextualization of findings (starting line 300).

Thank you!

Line 403 misspelled “Gadgill”

Good catch, thank you!

Throughout the results and discussion section, please highlight the relative importance of the non-microbial model predictors (e.g., the results in Table S7). While this paper is focused on the impacts of the microbial community, it is important to also note that other model predictors like soil pH, MAP, and tree type were significant in most models.

Thank you for the suggestion. This is something we have extensively discussed while preparing

the manuscript. We came to agree that this is beyond the scope of our manuscript since those results are not novel and those findings are repetitive to list across the different microbial groups analyzed. We now make a general acknowledgment to these predictors in the text and cue the reader to explore these non-microbial predictors in more detail in the supplement. At lines 197-201, we now write:

“Here, we show that none of these non-microbial predictor variables are strongly multicollinear with microbiome composition and diversity (variance inflation values ≤ 5 in all models sensu ⁶⁰), but they are important predictors of tree growth in our study (see the supplementary tables referenced throughout the results section).”

Conclusions:

Line 419 misspelled “carbo”

Good catch, thank you!

Supplement:

I am missing in the Supplement the number of sequences recovered across sites. Were there vastly different recovery of sequences across plots?

This table has more than 1,000 lines, but we have now added this as a supplementary table with the raw sequence counts per plot (see Table S10). As with all large-density sampling campaigns, there is substantial difference in sequences per sample, but we rarified the samples for all analyses requiring rarefaction in the text and thoroughly explain this in the methods section. At line 677 we now write: “...(see Table S10 for raw sequence counts).”